# Influence of Fruit Load Regulation on Harvest and Postharvest Fruit Quality and Antioxidant-Related Parameters in Sweet Cherry (*Prunus avium* L.) cv. Regina Cultivated under Plastic Covers in Southern Chile

**DOI:** 10.3390/plants13162257

**Published:** 2024-08-14

**Authors:** Jorge González-Villagra, Cristóbal Palacios-Peralta, Ariel Muñoz-Alarcón, Marjorie Reyes-Díaz, Pamela Osorio, Alejandra Ribera-Fonseca

**Affiliations:** 1Departamento de Ciencias Agropecuarias y Acuícolas, Facultad de Recursos Naturales, Universidad Católica de Temuco, Temuco P.O. Box 15-D, Chile; jorge.gonzalez@uct.cl; 2Núcleo de Investigación en Producción Alimentaria, Facultad de Recursos Naturales, Universidad Católica de Temuco, Temuco P.O. Box 15-D, Chile; 3Escuela de Agronomía, Facultad de Ciencias, Ingeniería y Tecnología, Universidad Mayor, Temuco 4801043, Chile; 4Centro de Fruticultura, Facultad de Ciencias Agropecuarias y Medioambiente, Universidad de La Frontera, Temuco P.O. Box 54-D, Chile; ariel.munoz@ufrontera.cl; 5Departamento de Ciencias Químicas y Recursos Naturales, Facultad de Ingeniería y Ciencias, Universidad de La Frontera, Temuco P.O. Box 54-D, Chile; marjorie.reyes@ufrontera.cl; 6Center of Plant-Soil Interaction and Natural Resources Biotechnology, Scientific and Technological Bioresource Nucleus (BIOREN), Universidad de La Frontera, Temuco P.O. Box 54-D, Chile; 7Research, Development and Innovation Department, Exportadora Rancagua S.A.-Ranco Cherries, Route 5 South, 04000, km 80, Rancagua P.O. Box 576, Chile

**Keywords:** fruit size, total phenols, postharvest storage, pedicel condition, pitting

## Abstract

Plastic covers have been used to prevent environmental constraints negatively affecting sweet cherry production in Southern Chile. However, less information is available on agronomic practices and their effects on fruit quality in sweet cherry covered orchards. Thus, in this study, we evaluated the impact of fruit load regulation on cherries’ antioxidant-related parameters and the quality and condition at harvest and postharvest in sweet cherry (*Prunus avium*) cv. Regina that was cultivated under a plastic cover in Southern Chile. For this, four fruit load treatments were manually applied—(i) 100% fruit load (the control), (ii) 80% fruit load, (iii) 60% fruit load, and (iv) 40% fruit load—in a commercial sweet cherry orchard for two seasons (2021/2022 and 2022/2023). The results revealed that the yield and fruit load were not significantly different between the treatments. Interestingly, the 60% and 40% fruit loads increased the fresh weight, fruit size, and firmness (20.3%) compared to the control (the 100% fruit load) during both seasons. Likewise, the 60% and 40% fruit load treatments exhibited the highest fruit size distribution of 30 mm, while the 100 and 80% fruit load treatments showed the highest fruit distribution with fruit sizes between 28 mm and 24 mm. The total soluble solids (TSSs) did not vary among the fruit load treatments, while a significant increase was found in the titratable acidity (TA) in the 60 and 40% fruit load treatments during both seasons. No significant differences in antioxidant activity (AA) and total phenols (TPHs) among the treatments were observed during both seasons. Overall, the results revealed that the fruit load treatments, mainly 40%, increased the fruit weight and firmness and reduced pitting in fruits by 39.4% at postharvest. Thus, fruit thinning might be an important agronomical practice to regulate fruit load, positively affecting fruit quality at harvest and during postharvest storage in sweet cherry cv. Regina cultivated under a plastic cover. However, more biochemical and molecular studies are needed to elucidate the mechanism involved in this improvement.

## 1. Introduction

Sweet cherry (*Prunus avium* L.) is a highly valuable fruit with remarkable vitamin C content and great antioxidant properties that promote human health [1,2]. Sweet cherries are an economically important fruit crop in temperate climates, including Chile [3,4]. According to FAOSTAT [5], Chile is the third country to produce sweet cherries worldwide, after Turkey and the United States. Sweet cherries is the main fruit crop cultivated in Chile, reaching 63,500 ha and 415,000 t of fruits exported during 2022/2023 [6].

Traditionally, the sweet cherry production area is mainly located between the Libertador Bernardo O’Higgins and Biobío regions [6]. However, during the last five years, the sweet cherry production area has increased to Southern Chile including the La Araucanía, Los Ríos, and Los Lagos regions (37°35′–40°33′ S), where mid- and/or late-maturing cultivars are grown, mainly for the Chinese market [3,7,8]. Nevertheless, Southern Chile has several environmental constraints for sweet cherry production, such as spring frosts and rain events, which mainly occur close to fruit bloom and harvest, decreasing fruit quality and yield [9,10]. Thus, orchard plastic covers have become a technological strategy to prevent the deterioration of fruit quality in sweet cherries due to environmental constraints [11,12]. Currently, in Chile, about 15% of sweet cherry production comes from plastic covered orchards, which has increased during the last few years [3,13]. However, more information is needed on the fruit quality, antioxidant properties, and postharvest fruit quality of sweet cherries cultivated under plastic covers [9,13]. Although some studies have reported that plastic covers modified the fruit quality parameters, less is known about agronomical practices and their effects on sweet cherry production under plastic covers [7,14,15,16]. In this regard, Salvadores and Bastías [3] suggested that sweet cherry production under plastic covers needs to be adjusted to agronomic practices such as pruning, irrigation, mineral nutrition, and fruit thinning. Fruit thinning is one of the most important agronomical practices to regulate fruit load and production in the sweet cherry, which may improve the firmness, fruit size, color, and total soluble solids and reduce the incidence of physiological disorders like pitting, pebbling, and pedicel browning [17,18]. However, previous studies have shown that plastic covers induced a heterogeneity in fruit maturity, with a significant difference in the fruit produced in the upper canopy zone compared to that produced in the lower canopy zone [7,16]. Therefore, reducing the fruit load could partially resolve this issue, improving the quality of the sweet cherry fruit.

Commonly, the thinning intensity is defined after the first natural fall of the fruits, a period in which it is evaluated whether the existing leaf/fruit (F/L) ratio is adequate to achieve an optimal relationship between productivity and quality. Studies have shown that fruit load regulation (reduction) has improved the fruit quality parameters in several fruit crops, such as apple, nectarine, sweet cherry, and olive, among others [19,20,21]. However, no information is available on the effect of fruit load regulation on the quality of sweet cherry cultivated under plastic covers.

Therefore, this study aimed to evaluate the antioxidant-related parameters and overall quality of sweet cherry (cv. Regina) cultivated in Southern Chile under a plastic cover at harvest and during postharvest.

## 2. Results

### 2.1. Yield and Fruit Physical Quality Analysis at Harvest

In our study, no significant differences were found in fruit yield and load among the treatments during the 2022/2023 season (Figure 1A,B). Although non-significant, we observed a decreasing trend in the 80, 60, and 40% fruit load treatments compared to the 100% fruit load treatment. Concerning the fruit quality analysis, the sweet cherry fruits from the 60% and 40% treatments showed significantly higher fresh weights compared to the 100% and 80% fruit load treatments in both the lower and upper canopy zones during the 20221/2022 season (Figure 2). Meanwhile, no differences were observed in the fresh weight among the fruit load treatments during the 2022/2023 season, with the mean values ranging from 9.8 ± 0.76 to 10.84 ± 0.31 g per fruit and from 11.93 ± 0.59 to 12.34 ± 0.28 g per fruit in the lower and upper canopy zones, respectively. Regarding fruit size (the equatorial diameter), the 60% fruit load treatment showed the highest values compared with the other treatments in the lower and upper canopy zones during both seasons. Interestingly, the fruits from the 40% fruit load treatment were firmer in the lower and upper canopy zones during both seasons, about 20.3% firmer compared to the control fruit load treatment (100%) during the 2021/2022 season (Figure 2).

#### 2.1.1. Fruit Size at Harvest

The size of the sweet cherry fruit was, on average, 28 to 30 mm in both canopy zones during both seasons (Table 1). Meanwhile, no fruits were found in the 32 and >32 mm fruit size range in either canopy zone in the season 2021/2022. During the 2021/2022 season, the fruit from the 60% and 40% treatments in the upper canopy zone were, on average, larger than the fruit from the other treatments, exhibiting a fruit size in the 30 mm distribution. The size of the cherries from the 100% and 80% treatments from the lower and upper canopy zones ranged from 24 to 28 mm. Regarding the 2022/2023 season, 32 and >32 mm fruit sizes were found in both canopy zones. We observed that the fruit from the 60% treatment was 37.5% larger than the control (the 100% fruit load treatment). In the lower canopy zone, the 100% fruit load showed the highest fruit distribution in the 26 and 24 mm fruit sizes compared to the other fruit load treatments. Meanwhile, no differences were observed among the fruit load treatments in the 28 and 30 mm fruit sizes in the upper canopy zone (Table 1).

#### 2.1.2. Fruit Color Distribution at Harvest

During 2021/2022, the color of the sweet cherry fruit ranged from red mahogany to mahogany, with the largest percentage of fruit being classified as mahogany (Table 2). In the 2022/2023 season, the fruits were mainly classified into the mahogany and dark color categories. In 2021/2022, in both canopy zones, the sweet cherry fruits presented the same color distribution among the treatments with a higher fruit load (100% and 80%), mostly classified into the mahogany color category.

The 40% fruit load treatment had a larger amount of fruit classified into the red mahogany color in the lower and upper canopy sizes. Otherwise, the color of the sweet cherry fruit was mainly red mahogany and dark mahogany in the 2022/2023 season. A similar tendency was observed in the fruit color distribution among the fruit load treatments, where a higher percentage of fruit from the 100% treatment was classified into the mahogany category, while most of the cherries from the 40% treatment were classified into the red mahogany category (Table 2).

#### 2.1.3. Chemical Analysis and Antioxidant-Related Parameters at Harvest

The TSSs did not vary among the fruit load treatments during both seasons (Table 3). By contrast, a significant increase was found in the titratable acidity (TA) of the fruit from the 60 and 40% treatments, which was 10% higher than the TA of the fruit from the control treatment (the 100% fruit load) (Table 3). Concerning the antioxidant-related parameters, no significant differences were observed in antioxidant activity (AA) among the fruit load treatments during both seasons (2021/2022 and 2022/2023) (Figure 3A,C). A similar tendency was observed in the total phenols (TPHs), where no differences were observed among the fruit load treatments in the 2022/2023 season, with the exception of 2021/2022, where higher TPHs were observed in the fruit from the 60% treatment (Figure 3B,D).

### 2.2. Postharvest Fruit Quality and Disorders

We analyzed the fruit quality and disorders in the cherries after 30 days of cold storage at 0 °C. Our results revealed that during the 2021/2022 season, the fruit from the 80, 60, and 40% treatments had a 10% higher fresh weight than the fruit from the control treatment (the 100% fruit load) (Table 4). By contrast, in 2022/2023, there were no significant differences in the fresh weight of the fruit from the different treatments. Regarding the fruit firmness, the fruit from the 80% fruit load were firmer among the treatments, increasing 29.3% with respect to the 100% fruit load treatment during the 2021/2022 season. Meanwhile, the 60 and 40% fruit load treatments exhibited greater firmness levels (7.4% and 20%, respectively) compared to the 100% fruit load treatment during the 2022/2023 season. Regarding the TSSs, a significant increase was found in the 80 and 60% fruit load treatments during the 2021/2022 season. The titratable acidity was not significantly different between the treatments (Table 4). 

No significant differences were found among the treatments for green or brown pedicel conditions or lacking a pedicel, regardless of the season (Table 5). In addition, the fruit from 2021/2022 showed no signs of pitting after cold storage, whereas the fruit harvested in 2022/2023 developed signs of pitting, regardless of the load treatment. Nonetheless, compared to the control (the 100% fruit load) and the 80% treatment, the fruit from the 40% treatment showed less pitting, followed by the fruit from the 60% treatment (Table 5).

In 2021/2022, there was also no significant difference in orange peel disorder between the load treatments. However, the fruit harvested in 2022/2023 was affected, particularly cherries from the 80 and 40% fruit load treatments (Table 5).

## 3. Discussion

Plastic covers have been used as an agronomic/technological tool to cope with environmental constraints in sweet cherry production in rainy and cold zones such as Southern Chile [10,11,12,13]. However, it has been reported that plastic covers induced heterogeneity in fruit maturity, which could be mitigated by reducing the fruit load [7,16]. Nevertheless, the impact of fruit load reduction on fruit quality and antioxidant properties has not been explored in sweet cherries cultivated under plastic covers [3,11]. Therefore, this study aimed to evaluate the effects of fruit load regulation on cherries’ antioxidant-related parameters, quality, and condition in sweet cherry (*P. avium*) cv. Regina cultivated under plastic covers in Southern Chile. The results from this study showed that the fruit load treatments did not affect the fruit yield and load. By contrast, the fruit physical quality analysis showed that the fruit from the 60% and 40% treatments exhibited significantly higher fresh weights at harvest compared to the 100% and 80% treatments during the 2021/2022 season. A similar tendency was observed in the fruit size, where the fruit from the 60% treatment exhibited the highest value during both seasons. Our results agree with those previously published by Blanco et al. [22], who reported that a 33% fruit load treatment increases fruit size compared to a 100% treatment in sweet cherry cv. Prime Giant. Similar results were recently reported by Matteo et al. [23], who showed that a 50% fruit load treatment improves fruit weight and equatorial diameter in sweet cherry cv. Lapins. Likewise, von Bennewitz et al. [18] reported that fruit load increases fresh weight and equatorial diameter in sweet cherry cv. Lapins. Thus, these authors showed that a 50% fruit load treatment produces larger fruit (between 28 and 30 mm in size), which was also observed in our study. Interestingly, the fruit load regulation produced larger fruits (between 32 and >32 mm) during the 2022/2023 season. Kurlus et al. [24] showed that fruit thinning reduces the number of small and medium fruits (<26 mm) and increases the number of fruits with a diameter of more than 28 mm in sweet cherry cv. Regina. According to Matteo et al. [23], fruit thinning increases the ratio of leaf area to fruit number (LA:F), raising carbohydrate availability and improving fruit weight, fruit size, and postharvest storage. Thus, Usenik et al. [25] suggested an optimal LA:F of 98.9 cm^2^ fruit^−1^, and a 60% fruit load produced higher quality fruits in sweet cherry cv. Lapins. Interestingly, the fruit from the 60 and 40% fruit load treatments were also firmer (by about 20%) compared to the fruit from the 100% fruit load treatment during both seasons. Thus, the 60 and 40% fruit load treatments might improve cherry fruit firmness, considering that Chile exported 87% of sweet cherry production to the Chinese market, where cherries travel between 20 and 30 days in marine containers [26,27]. Usenik et al. [25] and Blanco et al. [22] showed that fruit thinning improves fruit color and accelerates the ripening stage. During the 2021/2022 season, the color of the sweet cherry fruits was classified between the red mahogany and mahogany categories, with the color of the cherries from the 100% and 80% fruit load treatments being classified into the mahogany category in a higher percentage than the fruit from the other treatments (Table 2). During the 2022/2023 season, a higher percentage of the sweet cherry fruit was mainly in the red mahogany category. According to Salvadores and Bastías [3], the fruit color results are contradictory in sweet cherries cultivated under plastic covers, depending on the sunlight availability.

Previous studies have reported that plastic coverings decrease the TSSs and TA [7,28,29]. Our study showed no significant differences in the total soluble solids among the fruit load treatments. By contrast, Palacios-Peralta et al. [16] reported higher TSSs in sweet cherry fruits from uncovered plants compared to covered plants. Similarly, Bustamante et al. [7] showed higher TSSs in fruits from uncovered sweet cherry plants cv. Regina. It has been reported that trees received lower sunlight as a result of the plastic covering, reducing the TSS biosynthesis in the fruits [30,31]. We observed that the 60 and 40% fruit load treatments slightly increased the TA in the sweet cherry fruits compared to the 100% fruit load treatment. Our results agree with those from Matteo et al. [23], who showed that a 50% fruit load did not affect the TSSs and TA in sweet cherries. Sweet cherry fruits are a good source of phenolic compounds with antioxidant activity, promoting human health [32]. We observed that the fruit load treatment did not affect the AA and TPHs, with the exception of TPHs, which were higher in 60% of the fruit load treatments during the 2021/2022 season. In a previous study, Palacios-Peralta et al. [16] reported no differences in the AA and TPHs in fruits between covered and uncovered sweet cherry plants cv. Regina. They also reported that cyaniding-3-rutinoside and peonidin-3-rutinoside were the predominant anthocyanins in sweet cherry fruits. Our results agree with Khalil et al. [33], who showed that fruit thinning did not affect AA and TPHs in *Muscadinia rotundifolia*. By contrast, Li et al. [34] showed that reducing the fruit load increases the total phenols and antioxidant activity in *Vitis vinifera*.

After 30 days of cold storage, the fruit load treatments significantly increased the fruit weight and firmness compared to the control in our study (the 100% fruit load), while no changes were observed in the pedicel and fruit conditions. Unlike in our study, Matteo et al. [23] reported that cherry fruit firmness was not affected by fruit load treatments, but positive effects were observed in pedicel and fruit conditions, such as a reduction in the fruits without a pedicel or with a brown pedicel, as well as pitting, pebbling, and bruising after 45 days at 0 °C of postharvest storage. Likewise, Zoffoli et al. [35] reported that fruit thinning decreased fruit pitting (23%) and decay (53%) compared to no fruit thinning in sweet cherry cv. Van plants. Therefore, reducing the fruit load might be an important approach to improve the fruit weight and firmness at harvest and postharvest in sweet cherries.

## 4. Materials and Methods

### 4.1. Experimental Site and Plant Material

The field experiments were carried out at the commercial fruit orchard “La Ponderosa” (40°52′58″ S; 72°50′15″ O), located in La Union, the Los Lagos region, Chile, during two consecutive seasons (2021/2022 and 2022/2023). The weather conditions during the experiment are shown in Table 6. The weather data were obtained from the “Palermo” automatic weather station (AWS), located 10 km from the experimental site (https://agrometeorologia.cl, accessed on 13 July 2024). The plant material was obtained from sweet cherry (*P. avium* L.) cv. Regina plants established in 2017 on Gisela 6 Rootstock, spaced at 4.5 × 2 m and trained in a central axis system. The orchard was covered with a high-density polyethylene (HDPE) (with a density of 160 g m^−2^, 90% of total light transmission, and 60–65% diffuse light transmission). The plastic cover was deployed (closed) from the anthesis to the harvest. A total of 36 sweet cherry trees were selected for the experiment based on uniform vigor, development, and phytosanitary conditions. The agronomical management of the orchard, such as irrigation, fertilization, and pest control, was performed according to the technical recommendations of Exportadora Rancagua S.A.-Ranco Cherries.

### 4.2. Treatments

Sweet cherry (*P. avium* L.) cv. Regina trees grown under field conditions were subjected to four fruit load treatments: 100% (the control), 80%, 60%, and 40% of the fruit load. Before imposing the treatments, the sweet cherry trees were characterized by fruit number, and the fruits were hand-thinned at the stage of stone hardening (end of November of each season), as described by Penzel et al. [36]. The fruits were thinned at the stone hardening stage to ensure the fruit number in the treatments and to avoid fruit losses by environmental conditions, which is usually conducted in sweet cherry orchards by farmers. When the fruits ripened, a total of 200 fruits per each tree were harvested (100 fruits from the upper zone of the canopy (h > 1.2 m) and 100 fruits from the lower zone (h < 1.2 m)). The fruits were placed in a 1 kg export clamshell stored in a portable refrigerator and transferred to the laboratory of the Universidad de La Frontera, Temuco, Chile. The fruit analysis (antioxidant properties, quality, and condition) was conducted 24 h after the harvest and 30 days after the harvest, considered as the postharvest period, where the fruits were maintained under real exportation conditions at 0 °C and 90% relative humidity, as described by Bustamante et al. [7], in a freezer in the Exportadora Rancagua Company, Rancagua, Chile. For the antioxidant-related parameters, the fruits were dipped in liquid nitrogen and stored at −80 °C until the biochemical analysis, whereas the quality and condition of the fruits were evaluated under ambient temperature (20 ± 1 °C). 

### 4.3. Yield and Fruit Quality Analysis

All the ripe fruits in the trees were harvested simultaneously and weighed and counted to determine the fruit yield and number by plant, respectively. The fruit yield was determined using a precision balance (Model BA 220B, Biobase Meihua Trading, Jinan, China). For the fruit quality analysis, 100 fruits (from the upper and lower independent zones) were used to determine the fruit weight, equatorial diameter, firmness, soluble solids (TSSs), titratable acidity (TA), and maturity index (MI). The fruit weight was determined using an analytical balance (ML-204, Mettler-Toledo, Plainview, NY, USA). The fruit size (the equatorial diameter), fruit size distribution, and firmness were measured using a texture meter caliper (FirmPro, HappyVolt, Santiago, Chile). The TSSs were analyzed using the fruit juice, which was obtained from macerating fruits with a thermo-compensated digital refractometer (ATAGO mod. PAL-BX I ACID F5) standardized with distilled water and expressed as º Brix. The TA was determined by the volumetric titration method with sodium hydroxide (0.1 N) using an automatic titrator (HANNA mod. HI-84532). The TA was expressed as malic acid (%). Finally, the MI was calculated as the ratio between the TSSs and TA. 

The fruit skin color was analyzed by using an export scale as a reference (ASOEX, Las Condes, Chile), which included the following colors: light red, red, red mahogany, mahogany, and dark mahogany [37]. After 30 days, the fruit quality and disorder parameters were determined. The pedicel conditions, fruit cracking, pitting, and orange peel were determined as fruit disorders according to Wang et al. [38]; these condition parameters were evaluated in a sample of 100 fruits to three repetitions per treatment. 

### 4.4. Analysis of Antioxidant-Related Parameters

The antioxidant activity (AA) and total phenol (TPH) content were determined in the sweet cherry fruits at harvest. For this, 0.1 g of a fresh fruit sample from 12 fruits was macerated with ethanol (80% ethanol) and centrifuged for 10 min at 13,000 rpm. Then, the supernatant was collected and used for the AA and TPH determinations. The AA was determined using the highly stable free radical 1.1-diphenyl-2- picrylhydrazyl (DPPH), following the Chinnici et al. [39] method with some modifications. The absorbance was measured at 515 nm with a spectrophotometer (2800 UV/VIS, UNICO, Union City, NJ, USA). The AA results were expressed in micromoles of Trolox Equivalents (TE) per gram of fresh weight (FW). The TPHs were determined using the Folin–Ciocalteau method [40], measuring the absorbance at 765 nm (2800 UV/VIS, UNICO, Union City, NJ, USA) and using gallic acid as the standard. The results were expressed as mg of gallic acid equivalents (GAE) per g of FW.

### 4.5. Experimental Design and Statistical Analysis

The experimental design of this study consisted of a completely randomized block design. The data were analyzed using linear mixed models with the Infostat^®^ software version 2017. Significantly different means were compared using Fisher’s LSD multiple comparison tests (*p* ≤ 0.05). In addition, the analyses of the color distribution and fruit size distribution were analyzed using a generalized linear mixed model with a binomial distribution. 

## 5. Conclusions

Our study showed that control of the fruit load did not change the yield in sweet cherry cv. Regina. However, a reduction in fruit load, mainly 60% and 40%, improved the fruit weight, fruit size, and firmness at harvest and the fruit weight and firmness during postharvest storage in sweet cherry cv. Regina cultivated under a plastic cover. On the other hand, the TSSs and the antioxidant-related parameters were unaffected by the fruit load treatments, while the acidity was higher in the fruit from the 40% fruit load treatment. Therefore, a 60% treatment might be an interesting tool to improve the fruit quality at harvest and postharvest in a sweet cherry cv. Regina orchard under a plastic cover, which is important considering that Chile is an exporter country. However, more biochemical and molecular studies are needed to elucidate the mechanism involved in this improvement.

## Figures and Tables

**Figure 1 plants-13-02257-f001:**
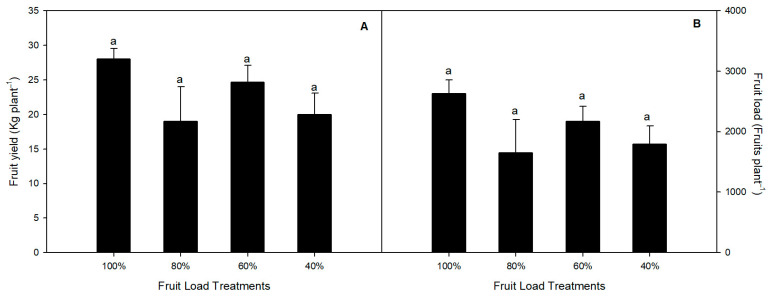
(**A**) The fruit yield and (**B**) fruit load at the harvest of sweet cherry (*P. avium*) cv. Regina subjected to different fruit load treatments during the 2022/2023 season. Different lowercase letters indicate significant differences among the fruit load treatments. The value represents the mean ± SE (*n* = 16).

**Figure 2 plants-13-02257-f002:**
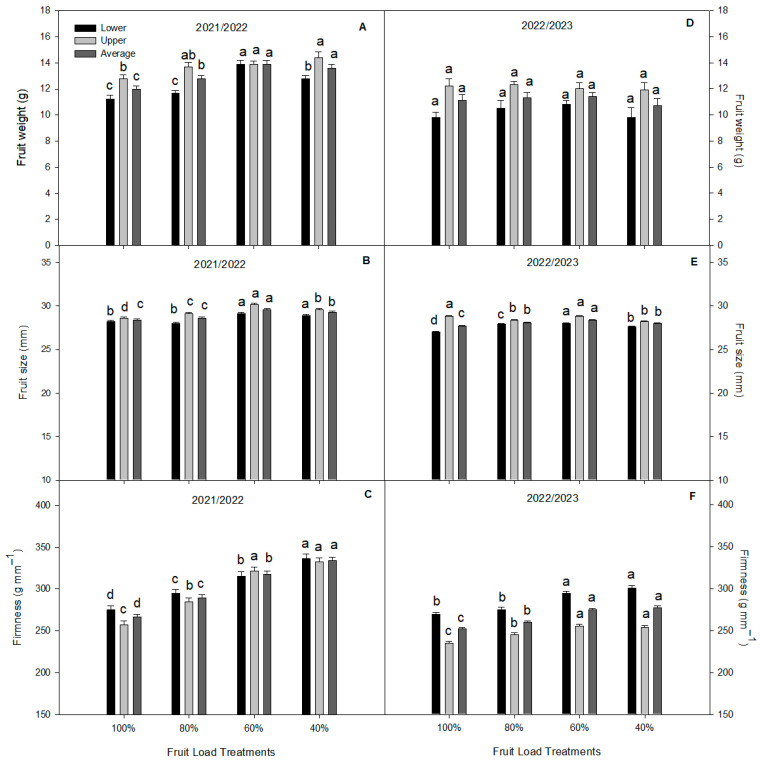
The physical fruit quality parameters at the harvest of sweet cherry (*P. avium*) cv. Regina subjected to different fruit load treatments during the 2021/2022 and 2022/2023 seasons. (**A**): Fruit weight, (**B**): Fruit size, and (**C**): Firmness at 2021/2022 season. (**D**): Fruit weight, (**E**): Fruit size, and (**F**): Firmness at 2022/2023 season Different lowercase letters indicate significant differences among the fruit load treatments for the same canopy zone and season. The value represents the mean ± SE (*n* = 100). Lower, Upper, and Average indicate the canopy zone of each tree calculated per treatment.

**Figure 3 plants-13-02257-f003:**
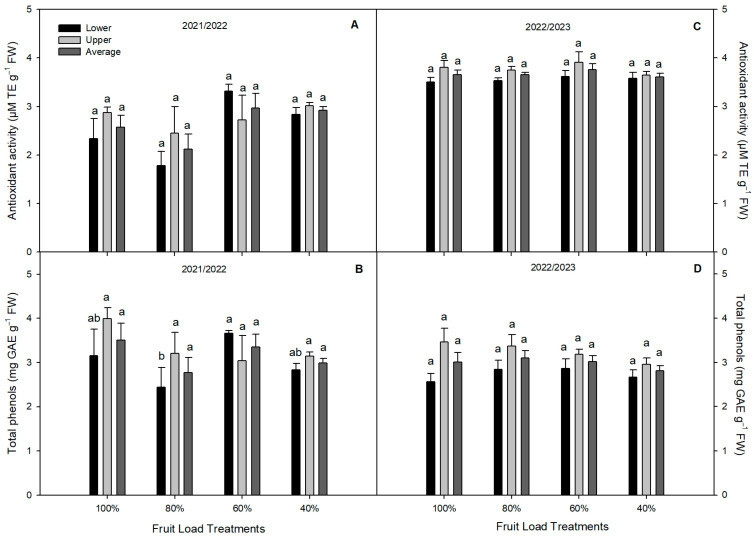
The antioxidant-related parameters in sweet cherry fruit (*P. avium*) cv. Regina subjected to different fruit load treatments during the 2021/2022 and 2022/2023 seasons. (**A**): Antioxidant activity, and (**B**): Total phenols at 2021/2022 season. (**C**): Antioxidant activity, and (**D**): Total phenols at 2022/2023 season. Different lowercase letters indicate significant differences among the fruit load treatments for the same canopy zone and season. The value represents the mean ± SE (*n* = 100). Lower, Upper, and Average indicate the canopy zone of each tree calculated per treatment.

**Table 1 plants-13-02257-t001:** Fruit size distribution at harvest of sweet cherry (*P. avium*) cv. Regina subjected to different fruit load treatments during 2021/2022 and 2022/2023 seasons.

Season	Canopy	Fruit Load	Fruit Size Distribution (%)					
	Zone	Treatment	24 mm	26 mm	28 mm	30 mm	32 mm	>32 mm
2021/2022	Lower	100%	5 ± 2 a	19 ± 3 a	49 ± 5 a	23 ± 5 c	n.d.	n.d.
		80%	2 ± 1 b	15 ± 3 ab	41 ± 5 a	39 ± 6 b	n.d.	n.d.
		60%	1 ± 0.3 b	10 ± 2 c	34 ± 4 b	55 ± 7 a	n.d.	n.d.
		40%	2 ± 1 b	12 ± 2bc	43 ± 5 a	40 ± 6 b	n.d.	n.d.
		*p*-value	0.0064	0.0148	0.0085	<0.0001	---	---
	Upper	100%	1 ± 0.1 a	13 ± 5 a	40 ± 6 a	41 ± 10 b	n.d.	n.d.
		80%	3 ± 0.1 a	10 ± 4 a	42 ± 6 a	40 ± 10 b	n.d.	n.d.
		60%	1 ± 0.1 a	6 ± 3 b	23 ± 5 b	67 ± 9 a	n.d.	n.d.
		40%	2 ± 0.1 a	6 ± 2 b	27 ± 5 b	64 ±10 a	n.d.	n.d.
		*p*-value	0.225	0.005	0.003	<0.001	---	---
	Average	100%	3 ± 0.1 a	17 ± 3 a	45 ± 4 a	32 ± 6 d	n.d.	n.d.
		80%	2 ± 0.1 ab	13 ± 3 a	42 ±4 a	39 ± 6 c	n.d.	n.d.
		60%	2 ± 0.1 ab	8 ± 2 b	28 ± 3 c	61 ± 6 a	n.d.	n.d.
		40%	1 ± 0.3 b	9 ± 2 b	35 ± 3 b	51 ± 6 b	n.d.	n.d.
		*p*-value	0.0216	<0.0001	<0.0001	<0.0001	---	---
2022/2023	Lower	100%	7 ± 1 a	20 ± 2 a	42 ± 3 a	25 ± 2 c	4 ± 1 b	0.48 ± 0.02 a
		80%	3 ± 1 bc	13 ± 2 b	31 ± 3 b	38 ± 2 ab	13 ± 3 a	0.47 ± 0.02 a
		60%	2 ± 1 c	10 ± 1 c	35 ± 3 b	40 ± 2 a	12 ± 2 a	1 ± 0.3 a
		40%	4 ± 1 ab	18 ± 2 a	31 ± 3 b	34 ± 2 b	11 ± 2 a	1 ± 0.4 a
		*p*-value	0.0004	<0.0001	0.0005	<0.0001	<0.0001	0.6991
	Upper	100%	2 ± 1 a	4 ± 1 b	25 ± 2 a	43 ± 3 a	21 ± 3 a	3 ± 1 a
		80%	1 ± 0.4 a	7 ± 2 b	28 ± 2 a	45 ± 2 a	16 ± 2 b	2 ± 1 a
		60%	1 ± 0.1 a	4 ± 1 b	23 ± 2 a	45 ± 2 a	22 ± 3 a	3 ±1 a
		40%	3 ± 1 a	11 ± 2 a	26 ± 2 a	42 ± 2 a	16 ± 2 b	1 ± 1 a
		*p*-value	0.2232	0.005	0.2563	0.5786	0.0141	0.1636
	Average	100%	5 ± 1 a	14 ± 1 a	35 ± 2 a	32 ± 2 b	12 ± 1 b	2 ± 0.45 a
		80%	2 ± 0.1 bc	11 ± 1 b	30 ± 2 b	41 ± 2 a	15 ± 1 a	1 ± 0.3 a
		60%	2 ± 0.1 c	8 ± 1 c	30 ± 2 b	42 ± 2 a	16 ±1 a	2 ± 0.45 a
		40%	3 ± 1 bc	14 ± 1 a	28 ± 2 b	39 ± 2 a	14 ± 1 ab	1 ± 0.3 a
		*p*-value	0.0001	<0.0001	0.0042	<0.0001	0.019	0.3929

Different lowercase letters indicate significant differences among the fruit load treatments for the same canopy zone and season. n.d. = not detected. The value represents the mean ± SE (*n* = 100). Lower, Upper, and Average indicate the canopy zone of each tree calculated per treatment.

**Table 2 plants-13-02257-t002:** Fruit color distribution at harvest of sweet cherry (*P. avium*) cv. Regina subjected to different fruit load treatments during 2021/2022 and 2022/2023 seasons.

Season	Canopy	Fruit Load Treatment	Fruit Color Distribution (%)				
	Zone		Red	Red Mahogany	Mahogany	Dark Mahogany	Black
2021/2022	Lower	100%	n.d.	32 ± 4 b	68 ± 4 a	0 ± 0 a	n.d.
		80%	n.d.	28 ± 4 b	70 ± 4 a	0 ± 0 a	n.d.
		60%	n.d.	34 ± 3 b	65 ± 4 a	0 ± 0 a	n.d.
		40%	n.d.	47 ± 3 a	53 ± 4 b	0 ± 0 a	n.d.
		*p*-value	--	0.0005	0.0009	n.s.	--
	Upper	100%	n.d.	27 ± 2 b	73 ± 2 a	0 ±0 a	n.d.
		80%	n.d.	22 ± 2 b	78 ± 2 a	0 ± 0 a	n.d.
		60%	n.d.	37 ± 2 a	62 ± 2 b	0 ± 0 a	n.d.
		40%	n.d.	40 ± 2 a	60 ± 2 b	0 ± 0 a	n.d.
		*p*-value	--	0.0004	0.0004	n.s.	--
	Average	100%	n.d.	30 ± 2 c	70 ± 2 a	n.d.	n.d.
		80%	n.d.	25 ± 2 d	74 ± 2 a	n.d.	n.d.
		60%	n.d.	36 ± 2 b	64 ± 2 b	n.d.	n.d.
		40%	n.d.	43 ± 2 a	56 ± 2 c	n.d.	n.d.
		*p*-value	--	<0.0001	<0.0001	--	--
2022/2023	Lower	100%	4 ± 1 b	17 ± 3 c	50 ± 3 a	26 ± 4 a	0.41 ± 0.0043 a
		80%	8 ± 1 a	25 ± 4 b	41 ± 4 b	23 ± 4 a	0.16 ± 0.0018 a
		60%	6 ± 1 ab	29 ± 4 b	49 ± 3 a	14 ± 3 b	0.17 ± 0.0019 a
		40%	5 ± 1 b	35 ± 5 a	40 ± 3 b	18 ± 3 b	0.14 ± 0.0016 a
		*p*-value	0.0345	<0.0001	0.0022	0.0001	n.s.
	Upper	100%	0.14 ± 0 a	6 ± 1 b	44 ± 6 a	48 ± 5 b	1 ± 0 a
		80%	0.08 ± 0 a	3 ± 1 c	36 ± 5 b	58 ± 5 a	1 ± 0 a
		60%	1 ± 01 a	13 ± 2 a	38 ± 5 b	46 ± 5 b	0.14 ± 0 a
		40%	0.2 ± 0.001 a	6 ± 1 b	36 ±5 b	56 ± 5 a	1 ± 0 a
		*p*-value	n.s.	<0.0001	0.0110	<0.0001	n.s.
	Average	100%	2 ± 0 a	11 ± 2 b	47 ± 3 a	37 ± 3 b	1.4 ± 0 a
		80%	4 ± 1 a	13 ± 2 b	39 ± 3 c	42 ± 3 a	1.0 ± 0 ab
		60%	4 ± 1 a	21 ± 2 a	42 ± 3 b	32 ±3 c	0.43 ± 0 c
		40%	4 ± 1 a	20 ± 2 a	38 ± 3 c	38 ±3 b	0.49 ± 0 bc
		*p*-value	0.0509	<0.0001	<0.0001	<0.0001	0.0246

Different lowercase letters indicate significant differences among the fruit load treatments for the same canopy zone and season. n.d. = not detected; n.s. = not significant. The value represents the mean ± SE (*n* = 400). Lower, Upper, and Average indicate the canopy zone of each tree calculated per treatment.

**Table 3 plants-13-02257-t003:** Fruit chemical quality parameters at harvest of sweet cherry (*P. avium*) cv. Regina subjected to different fruit load treatments during 2021/2022 and 2022/2023 seasons.

Chemical Quality Parameters	Season	Canopy	Fruit Load Treatments				
		Zone	100%	80%	60%	40%	*p*-Value
Soluble solids (Brix)	2021/2022	Lower	16.6 ± 0.41 a	15.7 ± 0.18 a	17.1 ± 0.34 a	16.6 ± 0.3 a	0.0734
		Upper	17.5 ± 0.29 a	17.6 ± 0.41 a	18.3 ± 0.29 a	18.1 ± 0.43 a	0.3725
		Average	17.1 ± 0.29 a	16.6 ± 0.41 a	17.7 ± 0.31 a	17.3 ± 0.37 a	0.1937
	2022/2023	Lower	16.4 ± 0.66 a	16.6 ± 0.39 a	16.4 ± 0.34 a	17.0 ± 0.28 a	0.5921
		Upper	18.5 ± 0.47 a	18.6 ± 0.25 a	18.9 ± 0.26 a	18.8 ± 0.22 a	0.722
		Average	17.4 ± 0.48 a	17.6 ± 0.34 a	17.7 ± 0.38 a	17.9 ± 0.29 a	0.8324
Titratable Acidity (% of malic acid)	2021/2022	Lower	0.36 ± 0.02 b	0.36 ± 0.01 b	0.41 ± 0.01 a	0.39 ± 0.01 ab	0.0391
		Upper	0.39 ± 0.02 b	0.41 ± 0.01 b	0.46 ± 0.02 a	0.43 ± 0.02 ab	0.0406
		Average	0.37 ± 0.01 b	0.38 ± 0.01 b	0.44 ± 0.02 a	0.41 ± 0.01 ab	0.0049
	2022/2023	Lower	0.49 ± 0.01 a	0.51 ± 0.02 a	0.52 ± 0.02 a	0.54 ± 0.02 a	0.1361
		Upper	0.52 ± 0.01 b	0.54 ± 0.01 ab	0.58 ± 0.01 a	0.56 ± 0.02 a	0.0278
		Average	0.50 ± 0.01 b	0.53 ± 0.01 ab	0.55 ± 0.01 a	0.55 ± 0.01 a	0.0083
Maturity Index (Brix/TA)	2021/2022	Lower	46.7 ± 2.03 a	43.8 ± 1.04 a	41.6 ± 1.13 a	43.08 ± 1.19 a	0.0846
		Upper	45.7 ± 2.26 a	43.1 ± 0.96 a	39.8 ± 1.58 a	42.29 ± 1.62 a	0.1524
		Average	46.2 ± 1.42 a	43.4 ± 0.67 b	40.7 ± 0.96 c	42.6 ± 0.94 bc	0.0025
	2022/2023	Lower	34.1 ± 2.05 a	32.9 ± 1.16 a	31.8 ± 0.86 a	31.9 ± 1.34 a	0.5942
		Upper	35.9 ± 1.15 a	34.5 ± 0.53 a	32.8 ± 0.61 a	33.8 ± 1.43 a	0.1228
		Average	35.0 ± 1.16 a	33.7 ± 0.65 a	32.3 ± 0.52 a	32.9 ± 0.98 a	0.106

Different lowercase letters indicate significant differences among the fruit load treatments for the same canopy zone and season. The value represents the mean ± SE (*n* = 100). Lower, Upper, and Average indicate the canopy zone of each tree calculated per treatment.

**Table 4 plants-13-02257-t004:** Quality parameters after storage, in sweet cherry (*P. avium*) cv. Regina from different fruit load treatments during 2021/2022 and 2022/2023 seasons.

Quality Parameters	Fruit Load Treatments	Season	
		2021/2022	2022/2023
Fruit Weight (g)	100%	10.8 ± 0.5 b	10.5 ± 0.4 a
	80%	12.1 ± 0.1 a	12.6 ± 0.9 a
	60%	11.9 ± 0.2 a	12.7 ± 0.5 a
	40%	11.9 ± 0.2 a	11.4 ± 0.8 a
	*p*-value	0.0402	0.1843
Fruit Firmness (g mm^−1^)	100%	309.0 ± 7.2 c	291.5 ± 4.8 b
	80%	437.1 ± 8.7 a	309.7 ± 13.0 b
	60%	325.5 ± 7.0 c	314.8 ± 23.7 ab
	40%	397.7 ± 7.7 b	360.6 ± 8.9 a
	*p*-value	<0.001	0.0413
Total Soluble Solids (Brix)	100%	14.9 ± 0.5 b	15.76 ± 0.3 a
	80%	16.9 ± 0.4 a	16.63 ± 0.2 a
	60%	17.8 ± 0.5 a	16.88 ± 0.5 a
	40%	16.3 ± 0.6 ab	17.11 ± 0.2 a
	*p*-value	0.04667	0.0775
Titratable Acidity (% of malic acid)	100%	0.33 ± 0.01 a	0.4 ± 0.03 a
	80%	0.27 ± 0.04 a	0.44 ± 0.03 a
	60%	0.26 ± 0.03 a	0.39 ± 0.06 a
	40%	0.38 ± 0.02 a	0.38 ± 0.02 a
	*p*-value	0.0877	0.6878

Different lowercase letters indicate significant differences among the fruit load treatments for the same canopy zone and season. The value represents the mean ± SE (*n* = 100).

**Table 5 plants-13-02257-t005:** Pedicel and fruit disorders after storage, in sweet cherry (*P. avium*) cv. Regina from different fruit load treatments during 2021/2022 and 2022/2023 seasons.

Season	Fruit Load	Pedicel Condition (%)			Fruit Condition (%)	
	Treatments	Green Pedicel	Brown Pedicel	Without Pedicel	Pitting	Orange Peel
2021/2022	100%	83 ± 2 a	14 ± 2 a	3 ± 0.01 a	n.d.	8 ± 2 a
	80%	87 ± 2 a	11 ± 2 a	1 ± 0.01 a	n.d.	11 ± 2 a
	60%	91 ± 2 a	9 ± 2 a	3 ± 0.01 a	n.d.	10 ± 2 a
	40%	85 ± 2 a	14 ± 2 a	1 ± 0.04 a	n.d.	9 ± 2 a
	*p*-value	0.0881	0.1933	0.1292	---	0.6495
2022/2023	100%	92 ± 2 a	4 ± 1 a	4 ± 2 a	38 ± 5 a	31 ± 5 ab
	80%	93 ± 2 a	2 ± 1 a	5 ± 2 a	37 ± 5 a	41 ± 5 a
	60%	89 ± 2 a	4 ± 1 a	7 ± 2 a	30 ± 5 ab	26 ± 4 b
	40%	95 ± 2 a	2 ± 1 a	3 ± 1 a	23 ± 4 b	39 ± 5 a
	*p*-value	0.1843	0.4781	0.2454	0.0277	0.0304

Different lowercase letters indicate significant differences among the fruit load treatments for the same canopy zone and season. n.d. = not detected. The value represents the mean ± SE (*n* = 100).

**Table 6 plants-13-02257-t006:** Weather variables during the experiment for both seasons (2021/2022 and 2022/2023) in La Unión, the Los Lagos region, Chile.

Season	Weather Variables	Month				
		October	November	December	January	February
2021/2022	Minimum Temperature (°C)	5.3	7.8	9.7	10.2	10.4
	Maximum Temperature (°C)	17.9	20.6	24.3	24.4	25.3
	Average Temperature (°C)	11.6	14.2	17.0	17.3	17.9
	Accumulated Rainfall (mm)	43.9	21.0	24.0	42.0	13.9
2022/2023	Minimum Temperature (°C)	5.1	9.0	9.6	10.2	9.6
	Maximum Temperature (°C)	17.2	22.4	23.5	25.5	25.8
	Average Temperature (°C)	11.1	15.7	16.5	17.8	17.7
	Accumulated Rainfall (mm)	32.3	16.8	20.4	17.6	7.2

## Data Availability

All data supporting the findings of this study are available within this paper.

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
