# Peer review of "Influence of Fruit Load Regulation on Harvest and Postharvest Fruit Quality and Antioxidant-Related Parameters in Sweet Cherry (Prunus avium L.) cv. Regina Cultivated under Plastic Covers in Southern Chile"

_plants, 2024, doi:10.3390/plants13162257_

Round 1

Reviewer 1 Report

Comments and Suggestions for Authors

This research is interesting, to control the fruit loading level is the most commonly recognized way to improve the quality of fruit. But how to control, control to what level is the fuzzy question that most operator confused. So this research have practical meaning. This paper have some place that need improve, the details are as below.

1. line 326, the load treatment is expressed as a retio, but how is ratio comes do not have a explain.

2. line 327, the treatment were conducted at the stage of stone hardening, why choose this stage? do we have any explain or references? usually the earlier to reduce the loading, the more to save the  nutrient substance.

3. how many trees are treated for each treatment ? 

Author Response

Dear reviewer,

Reviewer 2 Report

Comments and Suggestions for Authors

This MS studied the effect of fruit load regulation on cherries antioxidant related parameters, quality and condition at harvest and postharvest in sweet cherry. Below are the comments.

1.    Figure 1, though the decrease in fruit yield is not significant, it still may cause decrease in yield per acre.

2.    The image of the cherry load on trees and during storage may be shown.

3.    Line 172, mistake at “Figure 32A, C”.

4.    There are differences among parameters among different years. What caused this change?

Comments on the Quality of English Language

good

Author Response

Dear reviewer,

Reviewer 3 Report

Comments and Suggestions for Authors

Please see the file attached for suggestions, comments and editing.

Please add more details to the M&M section.

Comments on the Quality of English Language

English is fine but some sections would benifit from rewording, particularly the Results and Discussion sections.

Author Response

Dear reviewer,

Round 2

Reviewer 3 Report

Comments and Suggestions for Authors

Thank you for addressing my previous comments. I have only a few minor edits and suggestions. Please see the attached file.

Comments on the Quality of English Language

English requires some revision.

Author Response

Dear reviewer,
